# Recent Advances in MXene/Epoxy Composites: Trends and Prospects

**DOI:** 10.3390/polym14061170

**Published:** 2022-03-15

**Authors:** Raquel Giménez, Berna Serrano, Verónica San-Miguel, Juan Carlos Cabanelas

**Affiliations:** Department of Materials Science and Engineering and Chemical Engineering (IAAB), University of Carlos III of Madrid, Av. Universidad 30, Leganés, 28911 Madrid, Spain; rgimenez@ing.uc3m.es (R.G.); berna@ing.uc3m.es (B.S.)

**Keywords:** MXene, epoxy resin, MXene/epoxy resin composites, epoxy nanocomposites, polymer-hybrid composites

## Abstract

Epoxy resins are thermosets with interesting physicochemical properties for numerous engineering applications, and considerable efforts have been made to improve their performance by adding nanofillers to their formulations. MXenes are one of the most promising functional materials to use as nanofillers. They have attracted great interest due to their high electrical and thermal conductivity, hydrophilicity, high specific surface area and aspect ratio, and chemically active surface, compatible with a wide range of polymers. The use of MXenes as nanofillers in epoxy resins is incipient; nevertheless, the literature indicates a growing interest due to their good chemical compatibility and outstanding properties as composites, which widen the potential applications of epoxy resins. In this review, we report an overview of the recent progress in the development of MXene/epoxy nanocomposites and the contribution of nanofillers to the enhancement of properties. Particularly, their application for protective coatings (i.e., anticorrosive and friction and wear), electromagnetic-interference shielding, and composites is discussed. Finally, a discussion of the challenges in this topic is presented.

## 1. Introduction. Epoxy Resins and MXenes

As part of the thermosets family, epoxy resins (ERs) are suitable components for advanced engineering applications because they show outstanding mechanical properties and low shrinkage during curing, low residual stresses, and good thermal and chemical resistance [1,2]. They are also employed as the matrix of many fiber-reinforced polymers (FRP). ERs are usually composed of two components; a low-molecular-weight pre-polymer with two or more epoxide groups and a hardener or crosslinker, which can be an amine or anhydride compound or a catalyst. ERs offer a wide variety of combinations that make them suitable for a high number of applications, including adhesives, coatings, and composite materials [3].

Nevertheless, there is a high amount of interest in improving the performance of epoxy resins to give them other interesting properties. Like most polymers, epoxy resins are electrically non-conductive, and their thermal conductivity is also poor. Their good mechanical properties result from a highly crosslinked network, but this is also the reason why ERs are brittle materials with a moderate fracture toughness compared to other polymers. Many attempts have been made to overcome these drawbacks [4]. Blends with thermoplastics or elastomers improve the toughness, and inorganic particles such as silica or nano clays have been used. At present, the addition of nanoparticles shows great potential since their high surface-to-volume ratio enhances the interfacial aspects and thus requires low loadings. Two-dimensional nanomaterials are now very popular because of their outstanding properties derived from their high aspect ratio and specific surface area, surface chemistry, and quantum-size effects. Two-dimensional nanomaterials are excellent reinforcement phases for toughening and strengthening epoxy resins [5]. Graphene, a two-dimensional layer of carbon atoms sp^2^, is probably the most widely used 2D material. Beyond graphene [6], there is a huge abundance of two-dimensional materials; among them, hexagonal boron nitride (hBN) [7], transition metal dichalcogenides [8], and MXenes stand out in epoxy composites. Graphene has exceptional electrical and thermal conductivity. In addition, it has a high modulus and resistance. hBN has good thermal conductivity but is an electrical insulator. One of the most used dichalcogenides is MoS_2_, which shows dielectric properties and can be chemically modified. Dielectric nanofillers may be used as shielding against electromagnetic interference and microwave absorption.

Since their early discovery in 2011 [9], MXenes have been a fast-developing family of two-dimensional transition metal carbides, nitrides, and carbonitrides with unique properties [10]. MXenes can be described as *n* + 1 layers of a transition metal M covering *n* layers of X (where X is carbon or nitrogen). They are a 2D layered metal carbide and/or nitride with a graphene-like structure. Up to now, there are more than 30 different MXenes synthesized and reported. With general formula M_2_X, M_3_X_2_, and M_4_X_3_, they can be prepared from MAX phases (M_n+1_AX_n_) through chemical removal of the A layer (usually aluminum) and exfoliation (Figure 1). MXenes are also represented as M_n+1_X_n_T_x_. The T_x_ depicts the terminal groups in the surface, -O, -OH, or -F depending on the synthesis procedure. MXenes combine excellent metallic conductivity (up to 10,000 S cm^−1^ as a sheet) and high electrochemical activity. The hydrophilic character of the MXene sheets, due to the presence of hydroxyl or oxygen groups on their surface, allows a facile route for the preparation of stable delaminated aqueous dispersions. Moreover, they are biocompatible and show antibacterial properties. Owing to all those properties, they have been used for energy storage (batteries and supercapacitors) [11], environmental applications (the adsorption of pollutants, membrane separation, and desalination) [12], and EMI-shielding and radiation-absorption devices [13].

The literature on epoxy reinforced with nanofillers is extensive. Some recent reviews focus on using 2D nanofillers on polymers and epoxy resins [5,13,16], but they are either general or overly focused on one application. These publications scarcely mention several works with MXenes, partly owing to their novelty. However, the use of MXenes with epoxy resins is growing exponentially, and the literature indicates that knowledge advances rapidly. According to our analysis, publications are constantly increasing. For example, the number of publications in January of 2022 was the same as the number in the whole of 2018 (Figure 2a). The ratio of MXene/epoxy publications, as per testimonial 4 years ago, now reaches an appreciable 7% per each 100 graphene/epoxy publications and more than 66% concerning hBN/epoxy publications (Figure 2b).

In consequence, this review presents recent works about the use of MXenes as 2D nanofillers in epoxy resins. As shown below, MXenes enhance epoxy resins’ mechanical, thermal, and electrical properties and their fiber-reinforced composites. Applications in composites, flame retardancy, electromagnetic shielding devices, and anticorrosive/wear-resistant coatings have been reported and will be included (Figure 3). To increase their actual use, some MXene issues must be addressed. One of those issues is avoiding restacking the nanosheets, which negative impacts properties, even worsening the performance of the material. Other issues are mainly related to availability and cost and the stability of MXene nanosheets, which will also be considered in the review.

## 2. Preparation of MXene/Epoxy Composites

### 2.1. The Synthesis of MXenes

Two-dimensional structures can be obtained by separating the different stacked sheets constituting a compound, following a top-down method [10]. The procedure is based on the differences in the strength of the bonds within the sheets and between adjacent layers. If the bonds between layers are weak, which is usual, two-dimensional sheets can be obtained by breaking interlaminar forces. Numerous examples illustrate this concept. Graphene is obtained from graphite, and hexagonal boron nitride sheets [17] or molybdenum disulfide sheets [18] are also similarly obtained. Likewise, MXene sheets can be obtained from MAX phases. Bottom-up methods are also available to synthesize MXenes by chemical vapor deposition by template methods [19], but, up to now, the selected top-down method is well established and is more usual.

The top-down procedure involves two steps, etching and delamination. MAX phases are layered ternary carbides and nitrides with metallic and ceramic properties. The MAX phases, as stated above, have the structure M_n+1_AX_n_. To obtain the MXene sheets, the intermediate layer “A” atoms must be eliminated. The most common MAX phases are composed of Ti, Al, and C, in which aluminum is the element to be eliminated by etching. The etched layer is substituted by several T_x_ termination groups, which may be -OH, -O, or -F. Therefore, the material will consist of M_n+1_AX_n_ layers bonded by hydrogen bonds or other Van der Waals interactions. Ti_3_AlC_2_ is by far the most-used MAX phase, generating Ti_3_C_2_T_X_, but many others may also be etched to obtain different MXenes [20].

In order to remove the A-layer, fluorinated acid solutions are preferred. A HF solution is directly added [15,21,22] or formed in situ by reacting HCl with a fluoride salt (LiF) [23,24,25]. The MAX phase is stirred in these acid solutions. Concentration, reaction time, and temperature will affect the yield, and therefore, the structure of the MXene obtained [22,24,26].

After washing to remove the etched A-layer and the acids employed, it is necessary to ensure that the M_n+1_X_n_ sheets are adequately separated from the resulting multilayers, in which Li^+^ is usually introduced, increasing the interlayer space (Figure 1c). For complete delamination, several procedures are available. Direct sonication or stirring for a long time are valid options. Sonication helps to separate the nanosheets in shorter times. Another option is to use a solvent favoring the separation between sheets by intercalation, such as dimethyl sulfoxide (DMSO) or tetrabutylammonium hydroxide (TBAOH) [27]. MXene solutions can be used directly or by removing the solvent through filtration or freeze-drying [25,28,29]. Films obtained by vacuum filtration or spin-coating are common procedures to prepare adsorbents or electrodes, but they are not useful for epoxy resins. It is worth noting that in any case, the colloidal MXene suspensions must be used promptly or stabilized to avoid reaggregation and oxidation. Even more so because the M layer is more exposed after etching, and delaminated MXenes are susceptible to oxidation in water, generating TiO_2_.

### 2.2. Methods of Preparation of MXene/Epoxy Composites

The processing methods of epoxy resin nanocomposites with MXene nanosheets play an essential role in the final properties of the material. The selected fabrication technique will depend on incorporating MXene fillers with or without other additives. The chosen method should consider the versatility to integrate fillers, simplicity, ease of control, and facility to reach a homogeneous distribution of the fillers in the polymer matrix. Table 1 summarizes the advantages and disadvantages of the main fabrication methods: physical blending, infiltration molding, and vacuum-assisted impregnation. The diagram of Figure 4 correlates the target properties of the MXene/epoxy resin nanocomposites enhanced by each corresponding method.

Physical blending is divided into two methods, solvent-assisted blending and mechanical mixing. Solvent-assisted blending is a versatile and straightforward method to incorporate MXene nanosheets and other additives into epoxy resins. First, fillers are dispersed within a proper solvent, and then the liquid epoxy resin is added to the mixture. For solvent selection, both the MXenes’ dispersibility and the solubility of epoxy resin in the solvent are considered. For the waterborne type, water is used as the universal solvent. Nevertheless, organic solvents (mainly acetone, DMF, and chloroform) have also been used to obtain highly delaminated non-oxidized Ti3C2Tx MXene dispersions [30]. The process can be accelerated by high-shear stirring or sonication, although this may affect the structure of MXene nanosheets. The solvent is evaporated under vacuum or atmospheric pressure; therefore, remaining solvent in the nanocomposite and possible restacking of MXene nanosheets could cause drawbacks for specific properties and subsequent applications. Mechanical mixing avoids using a solvent by dispersing the fillers in the resin by applying high shear forces. In general, fillers are not easily well-dispersed, and restacking phenomena emerge as one of the main disadvantages of this processing method. Three-dimensional pre-fabricated structures to be used as fillers in epoxy resins involve the employment of other methods to backfill the polymer matrix without destroying the original structures. Infiltration molding and vacuum-assisted impregnation allow the infiltration of epoxy resin into the assembled structures by pouring the liquid resin into the structures and accelerating the backfill of the resin through the vacuum, respectively. Compared to physical blending methods, a 3D network of nanofillers or well-ordered aligned structures could be fulfilled by these processing methods without compromising the expected properties.

Figure 4 shows the methods of MXenes’ decoration of fibers to prepare fiber-reinforced epoxy composites. Usually, fibers are introduced in a solution of the MXene, and the nanosheets are deposited by grafting or electrostatic self-assembly.

## 3. Properties and Applications of MXene/Epoxy Composites

### 3.1. Mechanical Properties

The use of 2D materials, especially graphene, for epoxy resin reinforcement has been extensively documented. Important improvements in modulus, mechanical strength, toughness, and hardness have been reported, and the reader may consult interesting reviews [5,16,31,32]. In general, 2D nanomaterials have been studied in depth; however, the study of MXene-modified ERs is still in its beginning. The results vary greatly depending on how the MXene was obtained and dispersed, and the surface chemistry of the MXene nanosheets is also very relevant. Several early works have focused on how the MXene affects the mechanical performance of the epoxy matrix. The work by Wang et al. showed that the epoxy monomer itself could be intercalated and covalently linked to delaminated Ti_2_CT_x_ surface hydroxyls, providing improved mechanical performance through strong interfacial bonding [33]. SEM analysis showed that MXenes were mixed into the epoxy matrix as expanded bundles and single layers, and significant increases in impact strength and flexural strength were found for 1% composites. This is explained by the high stiffness of the MXene nanosheets, increasing the absorption energy and then enhancing their toughness. Additionally, crack deflection and shear yielding are facilitated by MXene, resulting in a rougher fracture surface. Higher concentrations have the opposite effect as excess Ti_2_CT_x_ introduced too many defects into the thermosetting network. Subsequent studies examined the influence of the MXene’s chemical nature and concentration (0.5 to 90%), its delamination procedures, and the method of introduction and dispersion (the use of a solvent, and mixing with sonication, high shearing, or a combination of both) into the epoxy resin. Micrographic analysis showed different dispersion states within the polymer matrix; therefore, the results are somewhat contradictory. Even so, all the research completed is coincident in that mechanical properties are usually enhanced to different extents, and fracture surfaces are always rougher than the brittle, fractured ER surfaces, showing a crack-deflection mechanism. Barsoum et al. [34] prepared MXene/ER composite by blending MXene with the epoxy precursor and found a limited mechanical behavior improvement in Ti_3_C_2_T_x_-reinforced ER. However, the MXene showed a partial exfoliation with a worse dispersion into the resin at the SEM scale.

Other authors used solvent-assisted methods to achieve suitable MXene dispersions into the matrix [25,35,36]. The work of Chen et al. revealed an improvement in both moduli (4.36 GPa, 20% modulus increase for 5% Ti_3_C_2_T_x_) and hardness by micromechanical analysis by nanoindentation [25]. The work of Gogotsi et al. is very interesting [35], as they prepared Ti_3_CN carbonitride MXene-modified ER at nanofiller ranges between 10% and 90%. The modulus was enhanced with MXene content up to three times the neat ER for the highest load, although the biggest percentage increments were found at lower loadings. Those authors established that MXene interaction with the epoxy was good enough to improve stress transfer and hinder crack propagation [25,35]. As the MXene content increased, the rigidity of the ER matrix chains could also increase through connection to the nanosheets [36]. Nevertheless, the different dispersion degrees led to different results. An excess of nanofiller worsened properties because more defects destroyed the network structure. The optimal nanosheets concentration was observed at 1% [33], 1.2% [36], and 5% [25] loadings, so it was not determined well, as it strongly depended on the delamination and dispersion state as well as the chemical nature of the MXene.

The recent works of G. Ying’s group focus on MXene surface functionalization to improve performance. Ti_3_C_2_T_x_ was chemically bonded with the hardener methyl tetrahydro phthalic anhydride (MTHPA), which could act as a disperse medium to avoid agglomeration or restacking of the MXene layers [37]. They found a notably tensile strength of 106.4 MPa (a +51% increment concerning neat ER) and flexural strength of 157 MPa (+32% with respect to ER) with only 0.2% Ti_3_C_2_T_x_. A 35% increase in the storage modulus and a decrease in T_g_ were observed for DMA with an increasing MXene content. These results are due to the reinforcing effect of the nanofiller bonded to the matrix network, enhancing the stiffness and slowing the T_g_ because of the impact over the crosslink density of the epoxy network. Attapulgite 1D nanorods (ATP, 20 nm diameter) were also recently employed to further functionalize the MTHPA-MXene surface by hydrogen bonding between hydroxyls of ATP and -COOH and hydroxyls from MTHPA-MXene (Figure 5a,b) [38]. Incorporating ATP nanorods increased the storage modulus, but an excess of ATP reduced the mechanical behavior because the ATP/MXene interface was weaker than ATP/ER or MXene/ER interfaces, and some ATP was dispersed into the ER. The best results were found for 0.2% MXene and 0.25% ATP, with increments in tensile strength and flexural strength of 88 and 57%, respectively, regarding ER. Notably, the stress intensity factor was also increased by 44%.

In fiber-reinforced polymers (FRPs), the challenge is to improve the interfacial interaction and, therefore, mechanical performance. Thus, MXenes have been introduced almost exclusively at the fiber–resin interface. The usual procedure is grafting or coating the MXene into the surface of the fibers. MXene-modified fibers are then put in contact with the ER. Usually, the interface shear strength (IFSS) or interlaminar shear strength (ILSS) is obtained from single-fiber pull-out or -fragmentation tests, but mat fiber composites have also been prepared. MXenes have been incorporated in aramid [4], carbon [29,39,40], and UHMWPE [41] fiber surfaces. A major common challenge of all FRPs is achieving a good interfacial strength between the fiber and the polymer matrix to control the load transfer under strain. Most FRPs use carbon fibers (CF), which have a hydrophobic and chemical inert surface. MXenes are hydrophilic, with abundant -OH and -O groups. Therefore, efforts may be made to improve their compatibility, modifying the MXene, the fiber surface, or both. Ying et al. [29] found effective functionalization of acid-treated CF with Ti_3_C_2_T_X_ through hydrogen interactions with carboxyl groups. They achieved a 186% improvement in interfacial shear strength. In addition, the surface topography of the modified CF changes, increasing the mechanical engagement effect during debonding. The same authors proposed incorporating short CF-functionalized MXene molecules into the ER matrix [42]. Activated SCF (around 20 μm) were immersed in a Ti_3_C_2_T_x_ dispersion. After coating, they were mixed with the ER by stirring under sonication. In this case, composites with a 2 wt % load showed the best behavior (tensile strength 100% higher than ER and 13% higher than a composite with non-coated SCF). MXene-coated SCF formed a stronger interfacial interaction with the epoxy matrix by physically interlocking and forming strong hydrogen bonds.

Sizing the CF with amino functionalities through aminopropyl triethoxysilane (APTES) grafting on previously acid-treated fibers is a common procedure to strengthen the interphase and has been essayed to deposit Ti_3_T_2_C_x_ through electrostatic interaction [39]. Tensile strength and flexural strength were increased by 40.8% and 45.9% with respect to CF-NH_2_/ER without MXene, and important improvements in impact strength were also observed. Ti_3_C_2_T_x_ nanosheets enhanced the interface connection between the fiber and ER and reduced interlaminar stress concentration because their mechanical properties are much better than the ER. Shur et al. essayed another route by grafting APTES to Ti_2_C nanosheets [40]. Ti_2_C-NH_2_ were then covalently grafted to the carboxyl groups of acid-treated CF through amide bonds. The remaining amine groups of the MXene could also react with the ER. Therefore, they play a bridge role in creating a strong chemical linkage between the fiber and matrix to impede debonding when the fiber breaks into shorter units as the stress transfer is improved, preventing crack propagation along with the interface. With respect to untreated CF, fewer interface failures were observed. Therefore, IFSS and ILSS were enhanced by 78% and 28%, respectively. Recent works have introduced additional components to the ER-MXene-CF system. In the work of Ao et al., MXene layers were electrostatically deposited in CF previously cationized with polyethyleneimine and then dipped into amine-modified SiO_2_ (Figure 5c) [43]. They found a significant increase in the surface energy of the sized CF, both for the polar component due to the functional MXene and silica and the dispersive component due to the wrinkled coverage with SiO_2_ nanoparticles, which increased roughness, enhancing wettability (Figure 5d,e). The high modulus of MXene caused the strength and stiffness of the interface phase, but, in addition, the rough surface provided a strong mechanical interlocking between the fiber and matrix. Therefore, IFSS increased by 56% in CF-MXene/ER concerning CF/ER, but a further increment to 72.75 MPa (+73.2% with respect to CF/ER) was observed in SiO_2_-decorated CF-MXene fibers. From the same authors, hierarchical structures on CF surfaces by self-assembly of chitosan and MXene layers inspired by a nacre-like layered structure were reported [44]. The amino groups of chitosan and hydroxyls of MXene were likely to form hydrogen bonds, with similar results in terms of surface energy and the enhancement of mechanical properties. Higher roughness, and therefore wettability, and more effective interface stress transfer enhanced interfacial adhesion and thus mechanical behavior.

Recent work from Xu et al. applied MXene to UHMWPE fibers for electronic textiles applications [41]. Plasma pre-treated fibers were coated with amphiphilic bovine serum albumin (BSA) before dipping into the MXene solution. Peptide regions of the protein may bond chemically or by a hydrogen bond with -OH or -COOH groups of the plasma-treated UHMWPE fibers and with -OH belonging to MXene. IFSS (~3.29 MPa) increased 116% and 31.6% compared with pristine UHMWPE fiber and UHMWPE/BSA fiber, respectively. Fewer interface failures and more matrix microcracks were developed in composites, which demonstrated good interface adhesion and improved load transfer between the fiber and matrix.

Theoretical studies have also been carried out on ER-MXene composites, either by using the finite-element method or molecular dynamics simulations. Sliozberg et al. [45] performed a computational investigation employing density functional theory and coarse-grained molecular dynamics to evaluate the MXene/ER composites’ mechanical behavior under stress and observed good stress transfer at the interface and MXene aggregation at high loads (4%). The simulation suggested that the increase in the filler sheets’ lateral size (aspect ratio) improved the reinforcement. Other works used finite-element methods to simulate the influence of aspect ratio, volume fraction, flakes alignment [46,47], and graphene addition to MXene-ER systems [48]. All these works are related by the observation of a higher modulus and tensile strength with the increase in the aspect ratio and if the nanosheets are aligned.

The most relevant results in the mechanical properties of MXene/ER nanocomposites are summarized in Table 2.

### 3.2. Thermal Properties and Flame Retardancy

With the fast development of technology, there is an increasing interest in polymer composites with high thermal conductivity (TC) and dielectric permittivity for microelectronics, energy storage, and other applications in which a good heat-dissipation performance is desired. The thermal-management capacity of composite materials can also reduce fire risks and system failures of electronic packages. Epoxy resins (ER) are widely used in the electronic fields due to their excellent mechanical properties, good chemical resistance, and high adhesion to substrates. Nevertheless, epoxy resins are characterized by having a very low thermal conductivity. Many attempts have been made to improve their TC by incorporating different inorganic nanoreinforcements, among them graphene/expanded graphite, alumina, and aluminum nitride. Inorganic nanomaterials may enhance both TC and fire safety. In recent years, different studies affirm that TC may be improved by incorporating MXenes to the ER, often reaching better results than other 2D nanofillers [28,49,50]. Most of the works focused on obtaining oriented composites to enhance thermal conductivity, so anisotropic conductivity was observed.

The addition of small amounts of MXene, between 0.2 and 1.0 wt %, improved the TC up to 141.3% concerning the pure ER [28]. This work also showed how the MXene nanofiller affected the TC of the resin with temperature. As the temperature rose, the segmental mobility of epoxy molecular chains increased, and therefore the mean free path of the phonons increased. The chains around the MXene filler were more ordered with the temperature increase and contributed to its enhancement. The MXene sheets also helped to decrease the thermal expansion coefficient.

MXene nanoflakes (multilayer) were introduced into the ER at much higher concentrations, between 5 and 40 wt % of reinforcement [49]. After mixing, the pre-cured composite fluid was slowly poured into preheated, slightly inclined shallow molds. These nanocomposites developed an anisotropic behavior. Therefore, two TC values could be obtained: in-plane TC and through-plane TC. In-plane TC measurements values were higher overall than through-plane TC; the in-plane TC increased by 505% (1.21 W/mK) for the sample with the lowest amount of MXene and 1470% (3.14 W/mK) for the sample with 30 wt %, while the through-plane TC increased by 13 and 47%, respectively. Results were attributed to the aspect ratio of Ti_3_C_2_T_x_ fillers and their aligned distribution in the epoxy matrix. In consequence, depending on the preparation method of the nanocomposite, the distribution of MXene sheets may be improved and increase the anisotropic conductivity. Lin et al. aligned the filler distribution by shear and centrifugal force in stirring by cast-moulding [50]. Due to the aligned distribution of the filler, the in-plane TC was improved, even for the samples with less MXene reinforcement, 6.45 times better than the same sample of 5% MXene without alignment.

The influence of MXenes on TC was also studied in carbon-fiber-reinforced ER. As it was mentioned before, CF is usually coated with the MXene nanosheets. However, in this work, a foam was prepared with MXenes (again in multilayer form), cellulose, and carbon fibers by solvent-mixing of the three components and oriented freeze-drying, and then the ER was infiltrated [51]. Therefore, the oriented columnar growth of ice crystals was used to obtain an oriented structure. MXenes were added at different degrees. TC was gradually improved as the concentration of MXene was increased. However, when the amount of CF-MXene increased, the TC passed a maximum and subsequently decreased. The TC of resins with CF-MXene reinforcements, 20% MXene and 30.2% of CF, was 9.68 W/mK. This was a 4509% improvement compared to epoxy resin and 36.7% better than resin with the same amount of CF without MXene. CF had an efficient synergistic effect with MXenes adhered by cellulose, which acted as a bridge between CFs. A very similar work also prepared oriented composites with 20 g of CF and different amounts of Ti_3_C_2_ (0.5–3 g) [52]. TC values reached 0.262 W/mK for the highest load. The epoxy composite is obviously amorphous, and the heat conduction mechanism was by phonons. The heat-conduction rate depended on the elastic mechanical properties of the material. The amorphous nature of ER scattered phonons severely; therefore, the TC of ER was very low (~0.2 W/m⋅K), but the oriented structure of the nanofiller created paths for conduction. When the temperature increased, epoxy segment mobility increased, enhancing the phonon path in the composite. On the contrary, the thermal expansion coefficient decreased when increasing the nanofiller. MXene and CF, oriented longitudinally, reduced the extent of volume expansion of the ER.

Interface thermal resistance between the filler and the ER may limit the TC performance. Therefore, the incorporation of MXene with silver particles was also studied. MXene/Ag powder was prepared by exfoliation, followed by a AgNO_3_ redox reaction [53]. Ag nanoparticles (AgNP) were then deposited on the exfoliated MXene to ensure good contact between nanosheets. MXene/Ag fillers clearly increased the TC of the resin. When the amount of mixed MXene/Ag nanofiller was 15.1% by volume in the resin, the in-plane TC was 1.79 W/mK and the through-plane TC was 2.65 W/mK, which is 1225% higher than neat epoxy. This result shows that Ag effectively improves the TC of the composites. MXene/silver aerogels act as heat-transferring skeletons for the epoxy nanocomposite. Following the same approach, Huang et al. [54] studied the TC of ER nanocomposites with Ag micro and nanoparticles, incorporating MXene or graphene nanosheets. ER/Ag composite showed a notable TC of 58.3 W/m⋅K. With an added MXene content as low as 0.12 vol %, the TC further increased to 72.7 W/m⋅K, which is 24.7% higher than the ER/Ag composite. The enhancement of TC was attributed to the combination of the bridging effect of AgNP particles via sintering and that of the MXene flakes via strong interactions with the AgNP. The MXene/Ag skeleton promoted the formation of heat-conduction paths in the matrix. Therefore, the combined use of MXene and Ag significantly improved the thermo-conducting properties of epoxy matrices and has been exploited in recent works with Ag nanoparticles [55] and using Ag nanowires (AgNW) [56]. Compared with pure epoxy, the 3D-oriented interconnection significantly improved the TC to 1.17 W/m⋅K in the MXene/AgNW/ER composites.

Flame retardancy is an interesting feature in a composite to increase fire safety. MXenes have been used to improve flame retardancy in ER. For this purpose, other components are usually added, such as red phosphorous (RP), a classic P-C-N system, zinc hydroxystannate (ZHS), or copper-organophosphate (CuP). MXene sheets were coated by RP through vapor exposition and calcination. RP-MXene was then introduced in the ER, showing excellent results in fire behavior tests [57]. RP is a well-known effective flame retardant, and RP-MXene acts as a physical barrier since it has a high specific surface and two-dimensional layers. Therefore, both P and MXene show a synergistic fire-retardant effect. In other works, waterborne ER was mixed with the P-C-N system (polyphosphate–pentaerythritol–melamine) and MXene flakes by blending [58]. Ti_3_C_2_T_x_ nanosheets enhanced the interfacial interaction between the carbonized layer and the ER, preventing the dropping of molten products during burning. It also strengthened the formed foam structure, preventing noncombustible gas release. In consequence, fire performance was improved. Song et al. [59] decorated MXenes with CuP, and Wang et al. decorated MXenes with ZHS [60]. The CuP-MXene composites showed good antibacterial and mechanical properties in addition to satisfactory flame retardancy. In ZHS-MXene composites, the epoxy was not damaged by fire because the nanosheets acted as barriers. Fire safety was improved due to the different routes built by the MXene and the free-radical-quenching capacity of the ZHS. In addition, the different MXene-forming pathways markedly reduced, by more than 50%, the maximum rate of heat release and the total heat released. The most relevant results in the thermal properties and flame retardancy of MXene/ER nanocomposites are summarized in Table 2.

We would like to briefly mention other properties that are included in the field of thermal properties, although they are not usually the target of the functionalization with MXenes. Numerous studies showed changes in the glass transition temperature of the ER when MXene nanofillers were added. The nanofiller may influence the T_g_ of a thermoset, but there is a confluence of several factors. First, the nanofiller disrupts the epoxy network; then, it can enhance mobility at the interface and decrease the T_g_. Second, the mobility at the interface is strongly dependent on the interfacial interactions between the nanofiller and ER matrix, so T_g_ may show an increase. Those interactions may be chemical, hydrogen bond, Van der Waals forces, or physical, for example, by interlocking the resin into a rougher surface. Generally, it was reported that the T_g_ increases as the amount of MXene into the composite increases because interfacial interactions are strong [61,62]. However, the effect is usually moderate, and increments of about 10 °C have commonly been reported. When the MXene amount is too high, the aggregation phenomena that reduce the MXene-ER interactions and the excessive increment of defects in the epoxy network led to a decrease in T_g_ [61]. Another interesting issue is to analyze is the thermal degradability of the ER when MXenes are part of the formulation. In this regard, there are contradictory results. Several studies affirm that the addition of MXene does not prevent degradation of the ER but effectively delays the initiation stage. Then, the degradation temperature increases as the amount of MXene in the samples increases [63]. By contrast, the samples with a higher content of MXene decreased their degradation temperature [35]. A slight improvement was observed at 5 wt % loading of Ti_3_CN, but composites with higher MXene content showed a decrease in degradation temperature. This decrease was attributed to the dispersion state of Ti_3_CN into the matrix. For high MXene loads, the amine-based hardener’s preferential adsorption could also affect the epoxy structure after curing.

### 3.3. Tribological Properties

Epoxy resins are components of adhesives, coatings, engineering plastics, and composites. Along with the mechanical and thermal behavior, wear resistance is very important to increase the lifetime of the industrial components. ERs are highly crosslinked and brittle; therefore, their tribological behavior is usually poor in extreme conditions, limiting their use in high-performance devices. Wear resistance and friction may be increased by adding micro or nanofillers (such as alumina or silica particles). Furthermore, properties such as hardness or impact strength are improved. Two-dimensional nanomaterials enhance the anticorrosive properties by increasing the corrosive ions pathway and providing good tribology. More traditional 2D materials such as boron nitride or graphene are chemically inert and therefore have poorer interaction with the matrix than MXenes, which, in addition, increase the wettability of the ER onto high-energy substrates. MXenes, therefore, are very good candidates to prepare high-performance anti-corrosive/wear coatings.

Several works introduced non-modified MXenes into ER coatings and investigated the tribological properties. Zhang et al. [52] prepared a three-dimensional Ti_3_C_2_ structure with carbon nanofibers through directional freezing and lyophilization, and then the ER was introduced by vacuum infiltration and cured (3 to 12 wt % MXene content). They observed an increase in hardness and thermal conductivity. Friction coefficient and wear rate decreased 76.3% and 67.3%, respectively, with the highest MXene load, regarding bare ER. The reduction in the wear rate was mainly ascribed to the excellent antifriction feature of Ti_3_C_2_ nanosheets. Ti_3_C_2_ -OH and -O groups allowed a good interface with ER and may participate in the curing reactions, enhancing the crosslinking at the interface. Under shear force, in high-load composites, the multi-layer Ti_3_C_2_ MXene presented was sheared to form a thin sheet, which made the shear strength in the friction process lower and thus had a particular effect on friction reduction. Few MXene sheets were on the CNF surfaces for low-load composites, but the friction coefficient was also reduced because the debris had a better self-lubricating effect. So, increasing the MXene amount does not directly translate to wear behavior. For example, more recently, Li et al. [62] prepared MXene/ER composites by solvent-blending. Increasing the amount of MXenes led to a progressive aggregation phenomenon, observed in the worn surface, which first slowed down the performance and even worsened when nanofiller content was around 2%.

Fan et al. functionalized MXene (Ti_3_C_2_T_x_) with APTES, taking advantage of their surface -OH [64]. Then, H_2_N-MXene was mixed (up to 0.5%) with waterborne ER and painted over Al alloys. Functionalization improved the stability of MXene-ER suspensions, and the mechanical properties were enhanced. The storage modulus increased, which brought about resisting plastic deformation and micro-cracking of the H_2_N-MXene/ER coatings. In addition, the better dispersion of functionalized MXene sheets prevented agglomeration and restacking, reducing stress concentration. The wear rate and friction coefficient decreased by 72.7% and 34% with respect to ER, and the improvement was clearly better when MXene sheets were covalently linked to ER through amino linkage. It must be noted that even when unfunctionalized, the interfacial adhesion between the polar MXene sheets and epoxides is high [65].

Recent works deal with incorporating nanoparticles to the MXene surface to improve behavior, favor dispersion, protect from oxidation, and avoid restacking of MXene layers. Wang et al. incorporated alumina particles [66] or PTFE nanoparticles (Figure 6a) [67]. PTFE had a very low friction coefficient, but its wear resistance was low, and it was not easy to disperse. Therefore, core–shell hybrids were prepared through electrostatic assembly. First, MXene was positively charged by modification with poly diallyl dimethylammonium, and methacrylate-modified core–shell PTFE latex was then deposited in the surface of the nanosheets.

MXene@PTFE hybrids were solvent-mixed with ER and cured. Friction and wear were studied at different environmental conditions. Compared to pure ER, the friction coefficient and wear resistance were highly improved. The friction coefficient was reduced by 14, 9, and 8.5 times, and wear rates were reduced by 29, 25, and 22 times in the air, humid, and vacuum conditions, respectively (Figure 6b,c). The electrostatic interaction inhibited the aggregation of the MXene sheets or the PTFE particles. Additionally, the modified MXene provided a stress-conduction channel and reduced the surface’s fatigue failure under friction. A stronger transfer film was formed, which reduced friction and wear. The modification with PDDA and PTFE also preserved MXene from oxidation.

Tang et al. incorporated ZrO_2_ nanodots to the MXene surface via hydrothermal synthesis, generating a high-density nanodots layer [68]. ZrO_2_-MXene was incorporated into the ER by solvent-mixing. The incorporation of the nanofiller significantly improved the wear rate of epoxy resin, and an optimal behavior was observed for a 0.5% ZrO_2_/Ti_3_C_2_ concentration. A minimum wear rate of 4.29 × 10^14^ m^3^/(N m) was obtained, reduced by 79.3% with respect to neat epoxy (Figure 6d). The storage modulus and T_g_ were also increased due to the strong interfacial interactions of -OH or -F groups with the epoxy resin, which locally restricted its molecular motion. Along with the good interfacial bonding, the worn surfaces and wear debris gradually changed from fatigue wear of neat ER to abrasive wear of 1% ZrO_2_-Ti_3_C_2_/epoxy composite. Similar results were obtained with TiO_2_ nanodots [61], with an optimum concentration of TiO_2_/MXene in the composite of 0.5%. In this work, the influence of the decoration degree of the nanosheets by TiO_2_ nanodots was analyzed, regulating the concentration of the TiO_2_ precursor solution. Low and mainly medium coverages showed better performance than high coverages. This was explained in terms of the TiO_2_ nanodot protuberances on Ti_3_C_2_, which induced a strong mechanical interlocking effect with the epoxy matrix, which can bear higher normal shear stress during sliding friction. If the coverage is very high, the interaction between TiO_2_ nanodots and the matrix is reduced (Figure 6e). Xue et al. focused their research on Ti_3_C_2_ as a solid lubricant [69]. In this work, the concentration of MXene was much higher than in the rest of the published work, from 30% to 90% by weight. Blending with the ER was completed by solvent-assisted stirring, and sonication and dispersion were drop-casted over an Al alloy. For an optimized MXene-content of 70 wt %, the friction coefficient was reduced by 66% with respect to ER. The epoxy resin formed the tribofilm with good adhesion and lubricating performance as a functional binder. The most relevant results of the tribological properties of MXene/ER nanocomposites are summarized in Table 2.

### 3.4. Electromagnetic Interference Shielding and Radiation Absorption

Electromagnetic interference (EMI) and microwave-absorption properties of 2D nanosheets, particularly MXenes, have attracted great interest due to their excellent electrical conductivity, high specific surface area, and chemically active surface. Electrical conductivity (σ) is especially a key factor in enhancing these properties when MXene is used as the filler of the matrix composite. Bearing in mind the low conductivity of epoxy resins, the addition of MXene nanosheets could remediate that handicap, leading to a significant improvement as the MXene loading increases. Recently, MXene/epoxy resin nanocomposites exhibiting good σ values have been reported [25,36,52]. Conductivity not only depends on the MXene amount added but also on the creation of conductive pathways in the composite. In that sense, oriented and interconnected layers increase conductivity. Nevertheless, the EMI-shielding performances of MXene/polymer composites are lower than those of MXene films, mainly due to the constraint in obtaining effective conductive pathways with distributed MXene and the high-volume fraction of the insulating polymer matrix. This paper reviews different MXene/epoxy resin nanocomposites that overcome those challenges.

The fabrication of hybrid absorbers is presented as an ideal alternative to using them as fillers for nanocomposites to improve microwave absorption. Hybrid absorbers offer the advantages of their individual constituents, such as nitrogen-doped graphene [70], carbon foam [71], or magnetic nanoparticles [72], and may display newly enhanced electromagnetic properties and performances for practical applications. Hybrid absorbers of 2 wt % nitrogen-doped graphene (N-GP) and up to 30 wt % Ti_3_C_2_ nanosheets were prepared by solvent-homogenized dispersion of the nanosheets with the epoxy resin [70]. The relative complex permittivity (real ε’ and imaginary ε’’ part) of the absorber can reveal interactions between electromagnetic waves in the microwave frequency band and the microwave-absorber-filled composites. The ε’ and ε’’ part of the hybrid absorbers were 54% and 35%, respectively, higher than those of the 40 wt % Ti_3_C_2_ nanosheets-filled epoxy composites due to their microstructure, larger internal boundary layer capacitance, microcapacity network formation, and dielectric properties. The potential use of epoxy-resin-based nanocomposites in the microwave-absorption field can be greatly enhanced by 3D MXene porous structures as absorbing fillers. Zhao et al. fabricated 3D Ti_3_C_2_T_x_ MXene (MX)/carbon foam (CF)/epoxy resin (ER) nanocomposites by adsorption of MXene on a carbon foam surface and freeze-drying, and subsequent vacuum-impregnation of epoxy resin [71]. The attenuation constant (α) determines the ability of the material to dissipate electromagnetic waves penetrating the composite and gradually increases with the MXene loading. When the MXene concentration was higher than 5 wt %, the attenuation constant decreased due to the decrease in the imaginary part (ε’’) of the relative complex permittivity. Figure 7a illustrates the proposed microwave-absorption mechanism of MX/CF/ER nanocomposites, mainly induced by the 3D porous network structure (inserted in the figure), surface functional groups, and defects. Decorated Ti_3_C_2_T_x_ MXene with magnetic nanoparticles to modify epoxy resin has recently emerged as an interesting strategy to enhance microwave-absorption performances [72]. Nanocomposites of Ni_0.6_Zn_0.4_Fe_2_O_4_/Ti_3_C_2_T_x_ (MP) were prepared by the hydrothermal method, and then they were mixed with epoxy resin to be used as a coating on a cement mortar specimen’s surface. The most excellent electromagnetic wave-absorption properties (EMW) were reached when 3 wt % MP was incorporated into epoxy resin, obtaining a reflectivity value 62.9% lower than that of the cement mortar specimen coated with a pure epoxy resin.

Nanocomposites of Ti_3_C_2_T_x_ MXene/epoxy resin, with the removal of partial polar groups on the surface of MXene by thermal annealing, which reached enhanced electrical conductivities and EMI-shielding performance, have been previously reported [25]. Nevertheless, in recent years, fabricating highly conductive polymer nanocomposites with good dispersion and the distribution of MXenes to form an optimal conductive network in the matrix has been a great challenge. The preparation of 3D MXenes/polymer EMI-shielding nanocomposites with low MXenes content, light weight, superior electrical conductivity, outstanding electromagnetic-interference-shielding performance, and excellent mechanical properties can be achieved. An effective strategy to improve the electromagnetic-shielding performance at low filler loading has been to use an aligned conductive network of nanofillers before impregnation with epoxy polymer. Aligned conductive networks can improve filler/polymer interface, enhancing the multiple reflections and reabsorption of electromagnetic waves between aligned fillers and improving the electromagnetic waves’ attenuation. Moreover, networks can decrease the percolation threshold of conductive fillers. In the following works, 3D hybrid porous structures were fabricated from Ti_3_C_2_T_x_ MXene with reduced graphene oxide [73], carbon foams [74], natural nanofibers [75], and silver nanowires [56] by directional-freezing and freeze-drying methods. Then, fillers were embedded in an epoxy matrix, providing the necessary strength and stabilization of the preformed aerogels, using a vacuum-assisted impregnation technique.

Zhao et al. reported the fabrication of epoxy nanocomposites with MXene/reduced graphene oxide (RGO) hybrid aerogels (MGA) obtained by hydrothermal reaction as fillers [73]. Conductive hybrid aerogels with an aligned cellular microstructure were constituted by MXene nanosheets as the shell of the cell walls and RGO sheets as the inner skeleton or core (Figure 7b). The nanocomposites’ electrical conductivity and EMI-shielding performances, with a Ti_3_C_2_T_x_ content as low as 0.74 vol %, presented a higher value than those of the previously reported polymer nanocomposites with graphene or other nanofillers. Likewise, 3D porous structures of Ti_3_C_2_T_x_/carbon hybrid foam (MCF), prepared by the sol–gel method, have been published [74]. With the addition of MXene, denser cells of MCF were achieved, and the contact between nanosheets gradually increased. MCF/epoxy resin EMI-shielding nanocomposites showed that electrical conductivity and the maximum total EMI-shielding effectiveness increased by 3.1 × 10^4^ times and 480%, respectively, compared with that of epoxy resin nanocomposites without hybrid foam.

Aerogels with aligned porous structures to be used as nanofillers in epoxy resin nanocomposites have been reported with other materials. On the one hand, cellulose nanofibers have broad application in EMI-shielding, energy storage, etc., besides being a green material with good biocompatibility and excellent mechanical properties. Annealed cellulose nanofibers/Ti_3_C_2_T_x_ MXene (TCTA) aerogels impregnated with epoxy resin showed excellent anisotropic electrical conductivity and EMI-shielding effectiveness [75]. The annealed TCTA/ER nanocomposites reached a percolation threshold of 0.20 vol %, much lower than that of other Ti_3_C_2_T_x_/polymer matrix nanocomposites. Absorption-shielding effectiveness increased drastically compared to reflection, demonstrating that absorption governed the shielding mechanism due to highly conductive porous networks and the persistence of aligned porous structures. STEM images (Figure 7c) evidenced that Ti_3_C_2_T_x_ structures’ integrity was well-maintained after impregnation with an epoxy matrix. On the other hand, 1D AgNWs are considered promising filler candidates for fabricating electromagnetic-interference-shielding materials due to their outstanding conductivity. Three-dimensional aerogels of MXene/silver nanowires (AgNWs) used to reinforce epoxy resin nanocomposites have just been reported [56]. In this publication, hybrid aerogels of MXene/AgNWs enhanced the EMI-shielding performance and remedied the typical weak adhesion between the cylindrical structure of AgNW and a substrate through hydrogen bond interactions with MXenes. Fabricated MXene/AgNWs/epoxy resin nanocomposite with 3 mm thickness showed the highest EMI shielding, 94.1 dB, enough to block practically 100% of the incident electromagnetic radiation and 79% higher than that of commercially standard EMI-shielding materials. Figure 7d shows these nanocomposites’ electromagnetic-waves-shielding mechanisms and shows the SEM image of MXene/AgNWs aerogel with 3 mm thickness.

Recently, another 3D hybrid porous structures/epoxy resin nanocomposite with superior EMI-shielding performance was reported. MXene nanosheets were assembled with honeycomb-structure reduced graphene oxide, first constructed using Al_2_O_3_ honeycomb as a template via electrostatic adsorption [76]. Next, honeycomb-structure rGO-MXene (rGMH)/ER nanocomposites were fabricated by pouring the epoxy resin and curing agent into the rGMH. The honeycomb structure (Figure 7e) ensured the distribution, shape, and control of cell size, extending the paths of electromagnetic waves to ensure superior EMI-shielding effectiveness at low MXene loading. Moreover, synergistic effects of MXene (3.3 wt %) and rGO (1.2 wt %) were translated into greatly improved electrical conductivity and EMI-shielding effectiveness, whose values were 2978 and 5 times, respectively, higher than those of the rGO-MXene/ER nanocomposites under the same fillers loading. The most relevant results in EMI-shielding and radiation absorption of MXene/ER nanocomposites are summarized in Table 2.

### 3.5. Anticorrosive Properties

Among organic coatings, epoxy resins (ER) are the most common approach for the corrosion protection of metallic structures [77]. Epoxy coatings provide a physical barrier effect to inhibit the penetration of corrosive media into metal surfaces. However, some micropores are easily created during solvent evaporation or/and the reaction of cross-linkers, which introduce channels for the diffusion of corrosive agents such as water molecules and Cl^−^ ions through coatings to the metal substrate. To overcome these problems, additives for acquiring high-efficiency corrosion protection are required.

Two-dimensional nanosheets have great potential to improve the barrier performance of epoxy coatings. Graphene [78], hBN [79], and transition metal disulfides [80] have been incorporated into epoxy formulations. The tortuous path created by well-dispersed 2D nanosheets and its impermeability to several gases and liquids agents gives them the ability to prevent and delay corrosive processes. MXene nanosheets can also form brick-wall nanostructures that significantly block species’ diffusion into the coating matrix. Due to this fact and the attractive properties of MXenes, their application in corrosion resistance has attracted research interest. Delaminated Ti_3_C_2_T_x_ has been the representative and most investigated MXene as an additive in epoxy coatings. This paper reviews different MXene/epoxy resin coatings to protect metal substrates from corrosion.

Based on the theoretical calculations, it has recently been reported [81] that Ti_3_C_2_T_x_ MXene, with F, O, and OH groups, has excellent adsorption stability for the epoxy group (and the benzene structures of DGEBA-based epoxy). Yan et al. [82] dispersed uniformly delaminated Ti_3_C_2_T_x_ in waterborne epoxy resin using a liquid-phase blending method. The presence of Ti_3_C_2_T_x_ at 0.5 and 1.0 wt % content inhibited the penetration of an electrolyte and considerably improved the corrosive resistance of epoxy coatings, which was attributed to the intrinsic properties of Ti_3_C_2_T_x_ and its strong barrier effect. However, the largest aggregates were detected with increasing content. The optimal addition of Ti_3_C_2_T_x_ in the epoxy coating was 1.0 wt %.

The dispersion degree of the MXene nanosheets significantly affects the anti-corrosion performance of coatings. In addition, similar to graphene, MXene may have the opposite effect due to the formation of electrodes by the highly conductive network in the coating matrix. Because of this, investigations have been focused on finding a method that could improve the dispersibility of MXene in the epoxy coating and inhibit galvanic corrosion. Yan et al. [64] synthesized an amino-functionalized Ti_3_C_2_T_x_, with 3-aminopropylthiethoxysilane (APTES), exhibiting good dispersibility and compatibility in the epoxy coating. Incorporating as low as a 0.5 wt % content of NH_2_-Ti_3_C_2_T_x_ sheets was enough to exhibit excellent anti-corrosion performance on an aluminum alloy substrate. This greatly improved performance was mainly attributed to the adhesion strength between the coating and metal substrate by the interaction between amino of APTES with hydroxyl groups on metal. Regarding water uptake, a significant contribution of amino-functionalized Ti_3_C_2_T_x_ sheets was observed. Impedance spectroscopy (EIS) in a 3.5 wt % NaCl aqueous solution was used to consider corrosion protection. The impedance modulus at low frequency (|Z|_0.01Hz_) from Bode-impedance plots is commonly used as a semi-quantitative indicator for assessing anticorrosion performance. Usually, the higher |Z|_0.01Hz_ represents better corrosion protection. For neat epoxy coating, the |Z|_0.01Hz_ value was 2.34 × 10^8^ Ω cm^2^ 24 h after initial immersion. After 4 weeks of immersion, the value decreased by three orders of magnitude, implying rapid coating destruction. When unfunctionalized MXene was incorporated (at 0.25 and 0.5 wt % content), significantly higher values after 4 weeks of immersion were reached compared to neat epoxy (about one order of magnitude higher). Meanwhile, the |Z|_0.01Hz_ values for amino-Ti_3_C_2_T_x_/epoxy were remarkably higher, especially at 0.5 wt % content (from 3.09 × 10^9^ Ω cm 24 h and 1.02 × 10^7^ Ω cm^2^ after 4 weeks), which was two orders of magnitude higher than neat epoxy coating, indicating excellent barrier properties. More recently, Li et al. [83] focused their study using (3-glycidyloxylpropyl) trimethoxysilane (GPS) to functionalize Ti_3_C_2_T_x_ MXene, with notable anticorrosion improvement from the better dispersibility and chemical bonding in the ER. The positive effect of epoxy functionalization on corrosion resistance was confirmed by EIS measurements using a 3.5 wt % NaCl solution as an immersion medium. The highest impedance was achieved for the 0.5 wt % MXene coating. In addition, it was further believed that GPS-Ti_3_C_2_T_x_ could repair the defects and compact the epoxy coating. In addition, the authors suggested that compared with amino-functionalized MXene (APTES-Ti_3_C_2_T_x_), the GPS-Ti_3_C_2_T_x_ nanosheet should be more suitable for practical application in epoxy coatings. Given the similarity between MXene and graphene, it makes sense to compare the behavior of both. Pourhashem et al. [84] found that GO functionalization with APTES provided superior enhancement in corrosion protection in an epoxy composite compared with a GPS-GO nanocomposite. Coated substrates with GPS-GO nanosheets displayed a higher contact angle and, therefore, higher coating resistance against water penetration than APTES-GO nanosheets. This phenomenon is only important in the initial immersion stage, where water absorption occurs. However, in the middle and final stages, barrier effects appear, and the dispersion and compatibility of the nanofiller become more important. Thus, the type of silane coupling agent applied significantly affects the corrosion performance, but it is unclear how it operates and what its role is, beyond improving dispersibility.

TEM images showed excellent dispersion and uniform GPS-Ti_3_C_2_T_x_ at low loading, but a change in GPS-Ti_3_C_2_T_x_ content from 0.3 to 0.7 wt % caused aggregation, weakening the barrier protection. A Nyquist diagram with a larger capacitive arc diameter corresponded to a better anticorrosion property. A significant reduction in capacitive semicircle diameter occurred for all coatings during immersion. This phenomenon is universal in EIS measurement. For neat ER, the smallest semicircles’ diameter in the Nyquist plots during the immersion stage are commonly observed along with the worst anticorrosion property. Unmodified Ti_3_C_2_T_x_ coatings only exhibited a slight improvement in Nyquist plots compared with neat ER. For the GPS-Ti_3_C_2_T_x_ coatings, the semicircle diameter increased when the amount was increased from 0.3 to 0.5 wt % but decreased significantly when the content was increased because aggregates were formed (Figure 8).

MXene functionalization is not always necessary. Taking advantage of the traditional cathodic protection of zinc, heterostructures were designed by grafting MXene (Ti_3_C_2_T_x_) on graphene oxide (GO) nanosheets without functionalization and incorporating them into an epoxy zinc-rich resin [85]. Previously, Ten S. et al. [86] already tested this concept with graphene. By open circuit potential (OCP) measurements, all coating systems studied (GO, Ti_3_C_2_T_x_, and GO-Ti_3_C_2_T_x_ zinc-rich epoxy coatings) remained within the scope of cathodic protection at the initial stage of immersion. However, coatings with Ti_3_C_2_T_x_ presented cathodic protection for a more extended period of immersion, especially for GO-Ti_3_C_2_ coatings, because they were able to provide electrical contact between zinc particles and the steel substrate, ensuring that sacrificial properties were not suppressed, which brought about a longer service life of the epoxy coating.

Zhao et al. [87] prepared noncovalent functionalized Ti_3_C_2_T_x_ MXene nanosheets with 1-(3-aminopropyl)-3-methylimidazolium bromide, an ionic liquid (IL), for waterborne epoxy coatings, to ensure superior oxidative stability of MXene in air. The IL simultaneously served as a high-efficiency corrosion inhibitor to prevent metals from corrosion and as a dispersant of Ti_3_C_2_T_x_ MXene into the epoxy coating. The electrochemical results confirmed an increase in the impedance modulus of the coating by 1–2 orders of magnitude higher relative to neat epoxy, containing small amounts of MXene. The superior anticorrosion performance of the composite coatings was mainly ascribed to the great increase in highly dispersed IL-MXene nanosheets and the extra property caused by the formation of IL passive films.

An inorganic–organic multilayer protection system consisting of Ti_3_C_2_T_x_/epoxy coating over a nitriding layer was used to protect aluminum alloy [88]. It is well known that nitriding treatment is an excellent approach to enhance the anticorrosion and antiwear properties of Al alloys. A nitriding layer with a thickness of about 120.8 μm was able to protect Al alloys, whereas the Ti_3_C_2_T_x_/epoxy coating kept the ability to repair the defects of the nitriding layer. Additionally, the impedance at low frequencies increased by two orders of magnitude after 4 weeks of immersion in a 3.5 wt % NaCl solution, confirming a significant advantage over the neat epoxy. The authors assigned this excellent performance to the barrier network formed by Ti_3_C_2_Tx and the shield of the nitriding layer.

Meng Cai et al. [89] designed Ti_3_C_2_Tx heterostructures with promising anticorrosion abilities using a (2D) layered double hydroxide (LDH) nanoclay based on Mg^2+^ and Al^3+^ assembled on surface MXene (MXene@LDH). Combining LDH antifriction performance as lubrication additives and, as recently reported, its active corrosion protection [90], the assembly of both nanosheets seems to be interesting, at least to resolve the inherent restacking of Ti_3_C_2_Tx MXene flakes. The MXene@LDH heterostructures exhibited excellent dispersibility and compatibility with the epoxy resin matrix. Comparing EIS measurements, larger capacitive arcs in Nyquist plots were observed for all fabricated coatings, LDH, MXene, and MXene@LDH epoxy composites, compared with neat epoxy. All formulations gradually deteriorated as the immersion time increased, but the MXene@LDH system displayed the largest radius in Nyquist curves for the different immersion times evaluated.

Electrical equivalent circuit models (Figure 8a) provide further theoretical interpretation for the corrosion process and are employed to fit the EIS data. In general, R_c_ (coating resistance) and R_ct_ (charge-transfer resistance) are related to the resistance of the penetration of electrolytes, and theoretically, the higher value of R_c_ and R_ct_ means good corrosion protection, i.e., a larger R_c_ represents better resistance of the coating. Thus, the R_c_ and R_ct_ values of all the coatings display a decreasing tendency with immersion time. Higher R_c_ and R_ct_ values compared to neat epoxy are commonly reported with the addition of MXene into the epoxy coating. However, important differences were observed depending on the type of functionalization used [64,83,89].

The work of Chen et al. [91] used silk fibroin (SF) to synthetize a SF-Ti_3_C_2_T_x_ hybrid as an additive for epoxy coating. Excellent anticorrosion behaviors under both atmospheric pressure and a simulated deep-sea environment were detected. A small amount of SF-Ti_3_C_2_T_x_ was sufficient to show an impedance value of 1.31 × 10^8^ Ω cm^2^, which is four orders of magnitude greater than pure ER. As a result of the MXene modification with SF fibers, the surface became rough, helping to improve the adhesion between Ti_3_C_2_T_X_ and the epoxy resin. The addition of 0.5 wt % SF-Ti_3_C_2_T_X_ was beneficial to achieve better corrosion resistance, even after immersion for 60 days at atmospheric pressure, and exceptional adhesion strength. Moreover, the |Z|_0.01Hz_ still maintained ~10^8^ Ω⋅cm^2^ after immersion for 240 h under severe hydrostatic pressure, offering ineffective failure prevention in harsh deep-sea environments.

Phosphorylated chitosan (p-CS) [92] and biodegradable amino acid L-Cysteine (Cy) [93] have also been used to functionalize MXenes, achieving good dispersion. The natural polymers improved the dispersion state and compatibility of MXene with the epoxy coating and inhibited galvanic corrosion, as shown by the mapping of coatings by EDS. The coating containing 0.5 wt % Cy-MXene nanosheets exhibited the best anti-corrosive properties. However, the best steel protected using p-CS-MXene/epoxy was the coating with 0.2 wt % content with minimal corrosion products, much better than with 0.5 wt % content. FTIR and XPS verified that modified chitosan interacted with MXene to form a Ti-O-P bond, contributing to optimal MXene delamination. The excellent uniform dispersion and distribution of CS-MXene into epoxy resin enhanced the barrier ability of the composite coating, making the penetration path of the electrolyte longer and more torturous, as well as helping to reduce the porosity of the coatings and the degree of delamination between the coating and substrate.

Novel strategies have also been reported. For example, Haddadi et al. [94] synthesized Ti_3_C_2_T_x_ MXene nanosheets modified with APTES and cerium cations (Ce^3+^) as an eco-friendly corrosion inhibitor to fabricate novel self-healing epoxy nanocomposite coatings. MXene-Ce^3+^ was introduced in the saline solution and into the epoxy resin and showed important improvement of the total impedance relative to epoxy, attributed to the release of Ce^3+^ cations into the corrosive medium, creating cerium-based components (cerium hydroxides/oxides), which act as a physical barrier for electrolyte diffusion, prolonging the durability of the coating. Another novel functionalization involving embedded carbon dots (CD) on the surface of MXene has recently been published [95]. A (CD)-MXene-epoxy for anticorrosion was prepared via Ti-O-C bonding and used to fabricate a self-aligned bioinspired CD-Ti_3_C_2_T_X_/epoxy composite compared with a randomly dispersed CD-Ti_3_C_2_T_X_/epoxy composite coating. The authors reported more than four orders of improvement in the impedance modulus because CD can easily be absorbed on the steel surface, providing an additional passivation effect. In addition, the parallel arrangement of the CD-Ti_3_C_2_T_x_ nanosheet can inhibit corrosion by eliminating connections of MXene/MXene and MXene/metal in the vertical direction, perpendicular to the diffused direction of corrosive species. An extraordinary R_c_ was reached, four and two orders of magnitude higher than that of neat ER and random CD-Ti_3_C_2_T_x_ coatings, respectively. In this work, the aligned CD-Ti_3_C_2_Tx epoxy coating showed the highest anti-corrosion performance reported, even compared to other forms of 2D nanosheet composite coatings such as graphene and BN-based epoxy composite coatings.

In summary, functionalized-MXene provides a promising method for designing nanocomposite coatings for high-performance and long-term metal protection. Nevertheless, despite the publications reported to date, it still remains a challenge to design a simple and effective process to fabricate MXene-based epoxy coatings with high anti-corrosion performances. More efficient protection strategies for MXene nanosheets are still to be explored. The most relevant results in the anticorrosive properties of MXene/ER nanocomposites are summarized in Table 2.

**Table 2 polymers-14-01170-t002:** Compilation of the most relevant results obtained with MXene/ER composites.

Filler *	Optimal Conc. (%)	Property	Performance ** (% Compared to Neat ER Except If Specified)	Ref.
Ti_2_C	1.0	Mechanical	IS: 17.8 kJ/m^2^ (+76%); FS: 98 MPa (+66%)	[33]
Ti_3_C_2_	5.0		Young modulus of 4.36 GPa (+20%), nanoindentation	[25]
Ti_3_CN	40–90		Young mod. of 12.8 GPa (+182%) for 90% Ti_3_CN, nanoindentation +93% in Young mod. and +104% in hardness for 40% Ti_3_CN	[35]
Ti_3_C_2_	1.2		IS: 24.2 kJ/m^2^ (+146%); TS: 66.2 MPa (+18%)	[36]
MTHPA-Ti_3_C_2_	0.2		TS: 106.4 MPa (+51%); FS: 157 MPa (+32%). MXene bonding to the matrix through MTHPA, which improves dispersion	[37]
ATP nanorods/MTHPA-Ti_3_C_2_	0.25/0.2		TS: 132.2 MPa (+88%); FS: 187.5 MPa (+57%). Covalent bonding between ATP and functionalized MXene	[38]
a-SCF/Ti_3_C_2_	2.0		TS: 141.2 MPa (+100%); FS: 199.3 MPa (+67%) Ti_3_C_2_ chemically bridges SCF and epoxy resin	[42]
a-CF/Ti_3_C_2_	1.37	Mechanical (Fiber-reinforced)	Single-fiber test. IFSS: 122.8 MPa (+182% compared to a-CF composite) Ti_3_C_2_ chemically bridges CF and epoxy resin	[29]
APTES-CF/Ti_3_C_2_	1.0		TS: 1210.9 MPa (+48.8% comp. to APTES-CF composite); FS: 987.3 MPa (+45.9% comp. to APTES-CF composite); IFSS: 223 MPa (+38.5% comp. to APTES-CF composite). Mxenes strongly attached to the NH_2_ functionality of APTES-CF	[39]
a-CF/APTES-Ti_2_C	0.2 mg/mL (DMF)		Single-fiber test. IFSS: 72.2 MPa (+78% compared to a-CF composite); ILSS: 44.2 MPa (+28% compared to a-CF composite). Amide bonding between a-CF and amino-functionalized Ti_2_C	[40]
PEI-CF/Ti_3_C_2_/APTES-SiO_2_	1 mg/mL (aq. Sol)		TS: 920 MPa (+26% compared to CF composite); FS: 1050 MPa (+39.2% compared to CF composite). Single-fiber test: IFSS: 72.75 MPa (+73.2%); ILSS: 78.7 MPa (+61.2%). Electrostatic assembly (positive PEI-CF/negative Ti_3_C_2_/positive APTES-SIO_2_). SiO_2_ enhances roughness and wettability with ER (+11% in IFSS and ILSS)	[43]
BSA-UHMWPE/Ti_2_C	1 mg/mL (aq. Sol)		Ti_2_C-decorated UHMWPE conductive fabric. IFSS: 3.29 MPa (+116% compared to UHMWPE composite)	[41]
Ti_3_C_2_	1.0	Thermal conductivity	TC: 0.587 W/mK (+141.3% compared to resin without additives)	[28]
Ti_3_C_2_T_x_	30		In-plane TC: 3.14 W/mK (+1470%); though-plane TC: 0.294 W/mK (+47%)	[49]
Ti_3_C_2_T_x_	40		In-plane TC: 1.29 W/mK (10.65 times better than ER); through-plane TC: 0.583 W/mK (2.92 times better than ER)	[50]
CF/Ti_3_C_2_	50.2		TC: 9.68 W/mK (4509% compared to ER and 36.7% compared to CF composite)	[51]
Ag/Ti_3_C_2_	15/0.1 (vol)		In-plane TC: 1.70 W/mK (+827%); through-plane TC: 2.65 W/mK (+1225%)	[53]
CF/Ti_3_C_2_	1.0		TC: 0.262 W/mK (+148% higher than ER)	[52]
Ag/Ti_3_C_2_	50/0.12		TC: 72.7 W/mK (+24.7% higher than ER with 50% Ag)	[54]
Ag/Ti_3_C_2_	1.0		TC: 0.382 W/mK (+135% higher than ER with Ag and +125% higher than ER with Ti_3_C_2_)	[55]
Ti_3_C_2_T_x_/AgNWs	4.1/4.1		TC: 2.34 W/mK (+1014% higher than ER and +200% higher than ER with Ti_3_C_2_T_x_)	[57]
RP-Ti_3_C_2_	2.0	Fire retardancy	LOI: improve values from 24.4 to 26.3% compared to ER	[57]
P-C-N/Ti_3_C_2_T_x_	3.0		LOI: increase 38% compared to ER	[58]
CuP-Ti_3_C_2_	5.0		PHRR: 64.4% less compared to neat epoxy	[59]
ZHS/Ti_3_C_2_T_x_	2.0		PHRR: 629.41 kW/m^2^ (−54.41% compared to neat epoxy)	[60]
Ti_3_C_2_	3	Tribological	COF: (−76.3%); WR: (−67.3%)	[52]
APTES-Ti_3_C_2_	0.5		COF: 0.357 (−34%); WR: 1.0 × 10^−13^ m^3^/(Nm) (−72.2%). Functionalization enhances dispersibility and reduces MXene amount	[64]
PTFE/PDDA-Ti_3_C_2_	2.0		Electrostatic interaction PTFE/PDDA-Ti_3_C_2_. PTFE inhibits aggregation. FC is reduced 8.5 to 14 times, and WR is reduced 22 to 29 times, depending on the environmental conditions	[67]
ZrO_2_/Ti_3_C_2_	0.5		COF: 0.6 (−35%); WR: 4.3 × 10^−14^ m^3^/(Nm) (−79.3%)	[68]
TiO_2_/Ti_3_C_2_	0.5		COF: 0.6 (−35%); WR: 3.3 × 10^−14^ m^3^/(Nm) (−84.5%). TiO_2_ nanodot protuberances induce mechanical interlocking effect	[61]
Ti_3_C_2_T_x_/LDH	0.5		WR was reduced by 80.45% compared to that of pure ER due to the synergy effect of MXene and layered LDH	[89]
Ti_3_C_2_T_x_/carbon foam	5 mg/mL (aq. Sol)	Radiation absorption	Porous structure allows incident EMWs to enter the material, and EM energy is dissipated through 3D MXene foam with high TC	[71]
Ni_0.6_Zn_0.4_Fe_2_O_4_/Ti_3_C_2_T_x_	3.0		EMW-absorption improvement of coatings for cement-based materials	[72]
Ti_3_C_2_T_x_/rGO	0.74 (vol)	EMI shielding	EMI SE: 56.4 dB (+210% compared to rGO composites)	[73]
Ti_3_C_2_T_x_/C hybrid foam	1.64/2.61		EMI SE: 46 dB and EC: 184 S/m (+480% and 3.1 × 10^4^ times, respectively, compared to C hybrid foam/ER)	[74]
Ti_3_C_2_T_x_/AgNWs	4.1/4.1		EMI SE: 94.1 dB (79% higher than the commercial materials)	[56]
Honeycomb rGO-Ti_3_C_2_T_x_	1.2–3.3		EMI SE: 55 dB and EC: 387.1 S/m (2978 and 5 times, respectively, compared to non-honeycomb-structured nanocomposites)	[76]
APTES-Ti_3_C_2_T_x_	0.5	Anticorrosive coating	|Z|_0.01Hz_ increments by 2 order of magnitude, due to good interaction between amino group of APTES and epoxy matrix	[64]
GPS-Ti_3_C_2_T_x_	0.5		|Z|_0.01Hz_ values 3 orders of magnitude. Good dispersibility because of the interaction between glycidyl groups of GPS and matrix	[83]
Ti_3_C_2_T_x_/GO	0.5		Superior corrosion resistance because of barrier effect. R_c_ values 1 order of magnitude higher than that of pure ZRC epoxy coating	[85]
Ti_3_C_2_T_x_/LDH	0.5		Good dispersibility and compatibility with ER. |Z|_0.01Hz_ values increase by 1 order of magnitude	[89]
SF-Ti_3_C_2_T_X_	0.5		Excellent dispersion in ER. Impedance value 4 orders of magnitude higher than pure ER after 240 h immersion (20 MPa pressure)	[91]
p-CS-Ti_3_C_2_T_x_	0.2		Uniform dispersion and distribution. Better compatibility with ER, reducing the porosity. |Z|_0.01Hz_ more than 2 orders of magnitude higher	[92]
CD-Ti_3_C_2_T_x_	0.5		With parallel arrangement, more than 4 orders improvement in impedance modulus, and Rc 4 and 2 orders of magnitude higher than pure ER and random CD-Ti_3_C_2_T_x_ coatings, respectively	[95]
Ti_3_C_2_T_x_	2.8	Self-healing	Self-healing was achieved in 10 s and 10 min under near-infrared and sunlight, respectively	[63]

* a-CF: activated carbon fibers; a-SCF: activated short carbon fibers; AgNWs: Ag nanowires; APTES: aminopropyl triethoxysilane; ATP: Attapulgite nanorods; BSA: Bovine serum albumin; CD: carbon dots; CF: carbon fiber; CuP: copper organophosphate; GPS: (3-glycidyloxypropyl) trimethoxy silane; LDH: layered double hydroxide; MTHPA: methyl tetrahydro phthalic anhydride; RP: red phosphorous; p-CS: phosphorylated chitosan; P-C-N: ammonium polyphosphate, dipentaerythritol and melamine; PDDA: polydiallyl dimethyl ammonium; PEI: polyethyleneimine; PTFE: polytetrafluoroethylene latex; SF: silk fibroin; UHMWPE: ultra-high-molecular-weight polyethylene fibers; ZHS: zinc hydroxystannate; ** COF: friction coefficient; EC: electrical conductivity; EMI SE: electromagnetic-interference shielding effectiveness; EMW: electromagnetic wave; FS: flexural strength; IFSS: interfacial shear strength; ILSS: interlaminar shear strength; IS: impact strength (toughness); LOI: limiting oxygen index; PHRR: peak heat release rate; TC: thermal conductivity; TS: tensile strength; WR: wear rate; |Z|_0.01Hz_: impedance modulus at low frequency from Bode diagrams.

### 3.6. Other Properties

#### 3.6.1. Self-Healing and Antibacterial Properties

The extraordinary physical and chemical properties of MXenes previously reviewed have attracted wide interest in the areas of mechanical engineering, optics, energy, and electronics. Currently, these materials are broadening their applications in the biomedical field. This is mainly due to their large surface area and strong absorbance in the near-infrared region. Furthermore, they present a highly efficient photothermal effect and more powerful antibacterial activity than other two-dimensional nanoparticles, such as graphene. Zou et al. reported the fabrication of multifunctional MXene/hyperbranched epoxy biomedical patches [96]. The hybrid patch was based on Ti_3_C_2_T_x_ MXenes, curcumin (cur)—which has multifunctional therapeutic properties—and epoxy-capped and phenylboronic-acid-decorated hyperbranched polyether (EHBPE-PBA), which was obtained through an epoxy/thiol curing chemistry. The resulting biomedical patch presented photo/thermal-induced self-healing ability and notable antibacterial activity owing to the high photothermal-conversion efficiency of Ti_3_C_2_T_x_ MXenes under near-infrared radiation at 980 nm. Furthermore, hybrid patches evidenced good surface wettability and water absorption thanks to the hydrophilicity of MXene nanosheets and the PEG-based polymer backbones forming EHBPE-PBA.

MXene nanosheets can absorb light and microwaves; hence, the thermally induced self-healing behavior of the coatings incorporating MXene was investigated. An epoxy coating, a reversible crosslinking network based on the Diels–Alder reaction, with dispersed MXene flakes, was fabricated [65]. MXene nanosheets generated heat by irradiating near-infrared light at 808 nm and focused sunlight to achieve the light-induced self-healing of the epoxy coatings. The self-healing properties of an epoxy coating with amino-functionalized Ti_3_C_2_T_x_ MXene loaded with 2-methylimidazole zinc salt (ZIF-8) nanocontainer@benzotriazole (BTA) multifunctional composite filler incorporated were also reported [97]. The coating exhibited efficient self-healing in a corrosive environment due to BTA promoting the formation of a coordination complex on the substrate surface. The possibility of chemically functionalizing MXene nanosheets’ surface with a self-healing agent such as benzotriazole contributed to homogeneously distributing the self-healing effect on the epoxy coating.

#### 3.6.2. Charge-Absorption Properties

MXene is a 2D nanomaterial with an extensive ion channel, allowing moderate charge adsorption capacity and a large specific surface area that is chemically very active. MXene/epoxy nanocomposites have been fabricated to eliminate the surface charge accumulation on insulators [98]. Epoxy resin was doped with different concentrations of Ti_3_C_2_T_x_ MXene, resulting in a doping level as low as 30 ppm, the most effective loading compared to those previously reported. The strong interaction between the functional groups and unsaturated bonds on the MXene surface and epoxy matrix in the interface region generated a high potential barrier under the action of an external field. Consequently, the energy was not enough to overcome the potential barrier, and most of the carriers were effectively trapped at the interface, suppressing the surface charge accumulation of the insulator.

## 4. Summary and Outlook

Epoxy resins are thermosets with interesting physicochemical properties for numerous engineering applications. They are also an ideal matrix of fiber-reinforced polymer composites. Considerable efforts have been made to improve their properties by adding nanofillers to their formulations. The selection of nanofillers has been marked not only by the intrinsic functional properties afforded to the material but also by their ease of synthesis, availability, and compatibility with the matrix. One of the most recent and most promising functional materials is MXenes, which can be obtained from MAX phases. MXenes are thermal and electrical conductors and show optimal mechanical performance. They are also hydrophilic with high-energy surfaces that are compatible with a wide range of polymers and may be functionalized/decorated with inorganic and organic compounds.

In epoxy resins, MXenes have been added with four main objectives: (i) to improve the mechanical performance of epoxy resins and fiber-reinforced epoxy composites, (ii) to increase the thermal and electrical conductivity of the resin, (iii) to improve the tribological and anti-corrosive behavior of epoxy resin coatings, and (iv) to improve radiation absorption for EMI shielding.

Despite the novelty of MXenes, considerable notable achievements have been attained on those objectives. From the point of view of mechanical properties, the introduction of MXenes into the epoxy resin improves the mechanical properties: the modulus, strength, and toughness. Nevertheless, a significant drawback of MXenes is the aggregation and re-stacking of the nanosheets for relatively high nanofiller loads, which limits their performance. The step of dispersion into the resin is, in fact, a fundamental stage to achieve good properties in the composite. In addition, MXenes must be manipulated with care and inert atmospheres at low temperatures to avoid oxidation. Recent efforts showed that the chemical surface groups of the delaminated MXenes can be functionalized with a variety of compounds and polymers to increase their compatibility with the resin, increase their dispersion, and protect MXenes from oxidation, with promising results. MXenes have also shown their usefulness in improving the behavior of fiber-reinforced epoxy composites. Taking advantage of their chemistry, MXenes have been used to modify the surface of activated carbon fibers, improving the fiber–matrix interfacial interaction. Remarkable increments in ILSS and IFSS have been reported. It should be noted that, although MXenes hold up well compared to other materials with similar structures, such as graphene, they provide additional interesting functionalities.

The improvement of the electrical and thermal conductivity of the epoxy resins by MXenes has been established. Notable electrical conductivities have been reported with low MXene loads, making their epoxy composites interesting for EMI shielding, anti-static coatings, and electronics and energy-related applications. The thermal conductivity of epoxy resins is highly increased by MXene addition. Especially in combination with Ag nanoparticles for reducing interface thermal resistance among the nanofiller and the epoxy matrix, MXene nanocomposites showed high conductivity and low thermal expansion coefficients. This effect widens the use of epoxy resins as thermal interface materials. The application of MXenes in epoxies for fire safety is now being explored. Preliminary works showed synergies when MXenes are combined with classic flame-retardant materials in the epoxy resin.

Epoxy resins are intrinsically brittle, so their tribological behavior is poor, a major drawback for their use as adhesives or protective coatings. MXenes have advantages with respect to traditional anticorrosive and antifriction nanofillers because their hydrophilicity enhances the interaction with the matrix and increases the epoxy resin’s wettability onto high-energy substrates. High-performance coatings with low friction and wear and/or anticorrosive properties have been prepared. For tribological performance, the best results are obtained when nanoparticles are introduced on the surface of MXenes. Those nanoparticles increase the roughness and favor mechanical interlocking with the matrix, help to reduce the friction coefficient, and may also protect MXenes from oxidation. Two-dimensional nanofillers have also been used to improve corrosion protection because of their barrier properties, which create tortuosity paths to delay the corrosion when microcracks or defects appear in the coating. MXenes show more important advantages: forming brick-wall structures that block the diffusion into the resin. A good dispersion of the MXene sheets is again fundamental to achieving optimal behavior. Recently, interesting results have been found with heterostructures created with MXenes and graphene or LDH nanoclays or by adding other components, reducing the contact between the MXene layers. Functionalized MXene introduction on epoxy coating formulations is undoubtedly a promising route to obtain high-performance anticorrosive coatings.

MXenes have acquired consideration in electromagnetic-interference-shielding and microwave-absorption fields for use as fillers, owing to their outstanding electrical conductivity, chemically active surface, and high specific surface area. The fabrication of Ti_3_C_2_T_x_ MXene/epoxy nanocomposites by blending may result in a simple and easily applicable method. Nevertheless, one of the drawbacks is the high MXene loading required to reach an effective conductive network. Synergistic effects in improving EMI-shielding and microwave-absorption performances have been provided by combining MXenes and other additives, such as magnetic nanoparticles, reduced graphene oxide, and silver nanowires. Fabrication of ordered structures with aligned conductive networks is an interesting strategy to improve conductivity at low filler loadings. We have identified recent works in which 3D hybrid conductive networks have been used as fillers of epoxy resin matrix exhibiting outstanding EMI-shielding properties. We consider this strategy a doubtlessly valuable methodology; nevertheless, it is not yet a significantly explored field. The freeze-drying fabrication process allows achieving porous conductive networks with low MXene content while maintaining a 3D structure. Despite this, MXenes are easy to oxidize, and EMI-shielding composite foam materials have poor air stability; therefore, the development of improved alternate fabrication methods continues to be a challenge.

In summary, MXenes have demonstrated that they may enhance a range of properties of great interest in epoxy resins. However, new challenges are still present that require addressing. The synthesis of MXenes has to be improved to reduce costs and favor their application. The oxidation of MXenes is an undesired possibility that has to be minimized by controlling the storage and application conditions. Functionalization will mitigate this drawback and help avoid the restacking of nanosheets. Many studies showed the potential of MXenes, the influence of the MXene’s chemical and physical structure on the composite performance remains unexplored. Indeed, most of the research on MXenes/epoxy nanocomposites has been performed with Ti_3_C_2_. Consequently, it is necessary to expand studies to other members of the extensive MXene family, not only carbides but also nitrides and carbonitrides. Moreover, except for a few papers, authors do not systematically analyze the influence of the size of the MXene sheets on the overall performance. The MXene dimensions may be a consequence of the delamination and dispersion procedure used, which may give rise to non-comparable results. A lack of standardization is also common in many nanomaterials and delays the research advances. At the current moment, anticorrosive coatings and EMI shielding appear to be applications with the most significant potential, but this may change if the availability of MXenes substantially increases. It is interesting to also note that, even the antibacterial and antifouling properties of MXenes begin to be established, there is a deficiency of studies related to this important feature on their epoxy nanocomposites with perspectives in environmental applications. This exhaustive review summarizes the previously reported properties exhibited by Ti_3_C_2_T_x_ MXene/epoxy resin nanocomposites. We are looking forward to providing insight into future trends and challenges for developing materials with enhanced properties for next-generation applications.

## Figures and Tables

**Figure 1 polymers-14-01170-f001:**
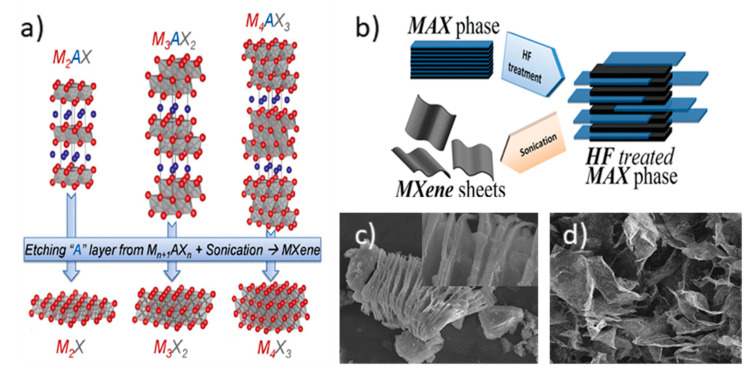
(**a**) Scheme of the obtention of different MXenes from their MAX phases. Reprinted with permission from Ref. [14]. Copyright 2014 Wiley; (**b**) Delamination procedure includes HF/HCl treatment to etch Al atoms, intercalation, and delamination by shear and/or sonication. Reprinted with permission from Ref. [15]. Copyright 2012 American Chemical Society; (**c**) Ti_3_AlC_2_ after acid treatment (accordion-like); (**d**) Ti_3_C_2_T_x_ MXene sheets thoroughly exfoliated (images from the review authors).

**Figure 2 polymers-14-01170-f002:**
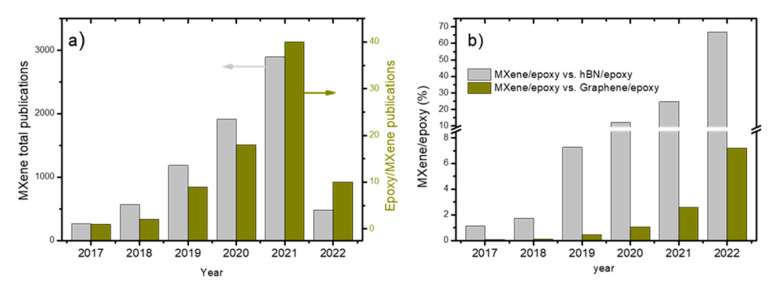
(**a**) Summary of publications of MXenes and MXenes and epoxy from 2017 to 2021; (**b**) Summary of MXene/epoxy publications (%) related to hBN/epoxy (grey) and graphene/epoxy (green) publications from 2017 to 2021. Data from 2022 are from January. Results from Web of Science (February 2022).

**Figure 3 polymers-14-01170-f003:**
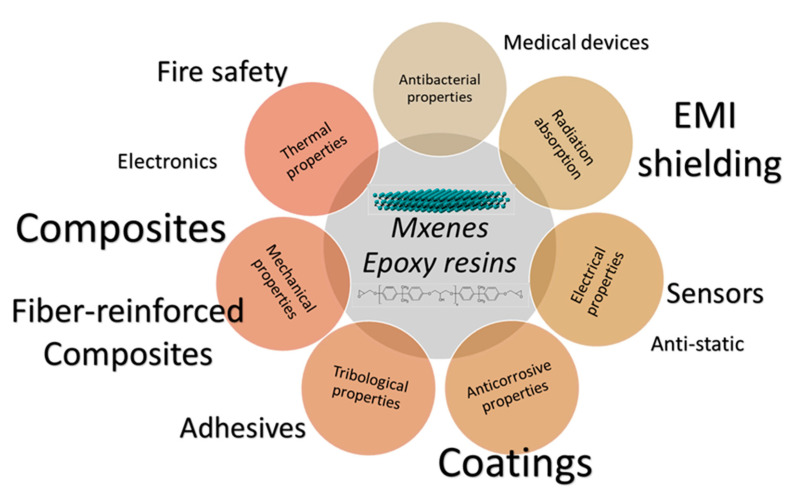
Overview of the applications of MXenes in epoxy resins.

**Figure 4 polymers-14-01170-f004:**
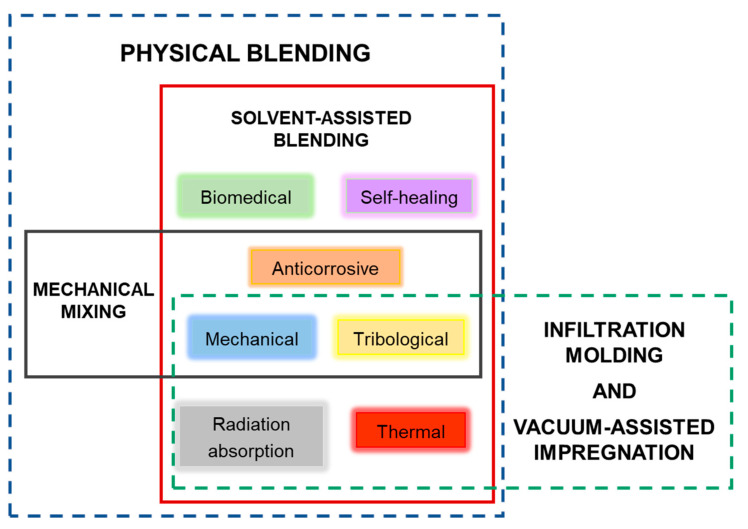
Diagram of processing methods of MXene/epoxy nanocomposites and related properties of nanocomposites fabricated by the corresponding method.

**Figure 5 polymers-14-01170-f005:**
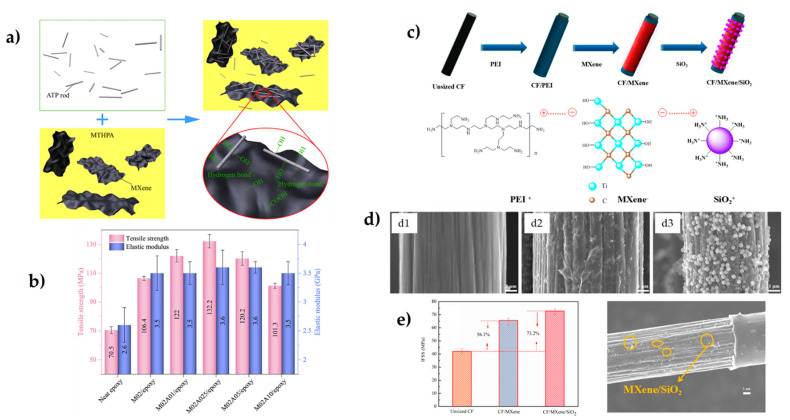
(**a**) Scheme of the preparation of Attapulgite (ATP)-MXene hybrids. Reprinted with permission from Ref. [38]. Copyright 2021 MDPI; (**b**) Tensile strength and elastic modulus for ATP-MXene/ER composites. Reprinted with permission from Ref. [38]. Copyright 2021 MDPI; (**c**) Scheme of the preparation of SiO_2_-decorated, MXene-modified carbon fibers. Reprinted with permission from Ref. [43]. Copyright 2022 Society of Plastic Engineers; (**d**) Surface morphologies of CF (d1), CF/MXene (d2), and CF/MXene/SiO_2_ (d3). Reprinted with permission from Ref. [43]. Copyright 2022 Society of Plastic Engineers; (**e**) IFSS of the composites and SEM image of fracture surface of CF/MXene/SiO_2_ after debonding test. Reprinted with permission from Ref. [43]. Copyright 2022 Society of Plastic Engineers.

**Figure 6 polymers-14-01170-f006:**
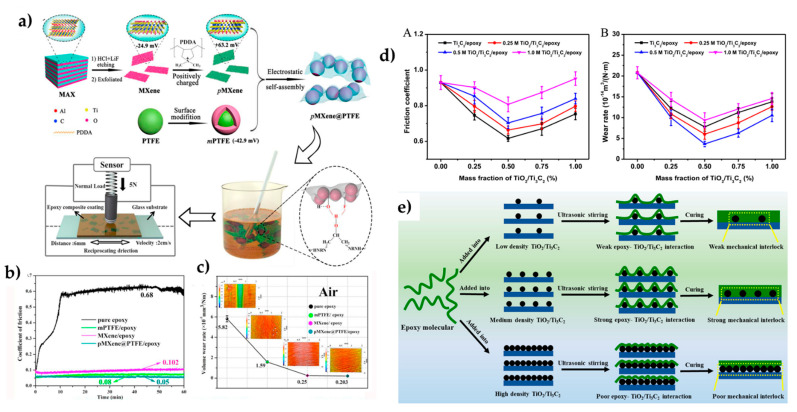
(**a**) Schematic illustrating of fabrication for MXene@PTFE hybrid, preparation process, and friction test of epoxy composite coating. Reproduced with permission from Ref. [67]. Copyright 2021 Elsevier; (**b**) COF curves and (**c**) volume wear rates (W) of pure epoxy, PTFE/epoxy, MXene/epoxy, and MXene@PTFE/epoxy composite coatings under humid conditions (RH: ~80%). Reproduced with permission from Ref. [67]. Copyright 2021 Elsevier; (**d**) Friction coefficient (A) and wear rate (B) of TiO_2_/Ti_3_C_2_/epoxy nanocomposites with different mass frictions under a normal load of 8 N. Reproduced with permission from Ref. [61]. Copyright 2021 MDPI. (**e**) Model of the interaction between TiO_2_/Ti_3_C_2_ and epoxy matrix in TiO_2_/Ti_3_C_2_/ER nanocomposites. Reproduced with permission from Ref. [61]. Copyright 2021 MDPI.

**Figure 7 polymers-14-01170-f007:**
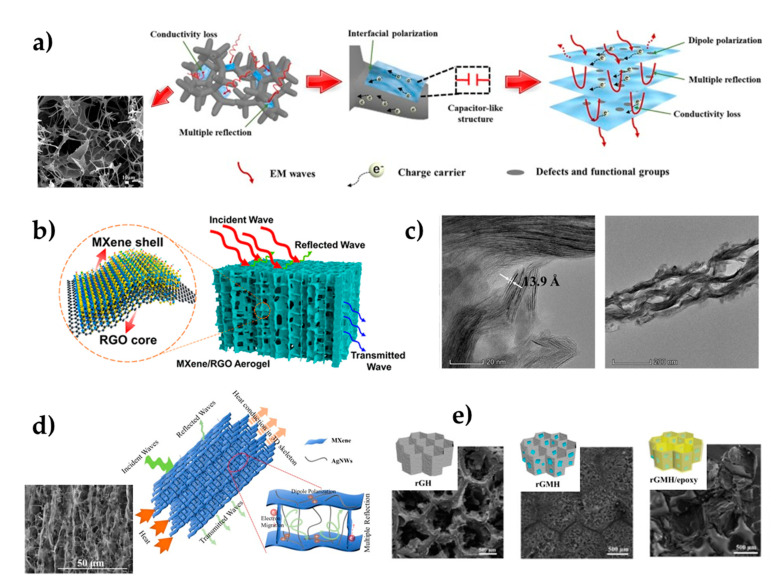
(**a**) Schematic diagram of the potential electromagnetic wave absorption mechanisms for the MS/CF/ER composites and SEM image of MX/CF foam at concentration of 5.0 mg/mL of Ti_3_C_2_Tx MXene. Reprinted with permission from Ref. [71]. Copyright 2021 Elsevier; (**b**) Schematic illustration of Ti_3_C_2_Tx MXene/RGO hybrid aerogel structure and EMI-shielding mechanism. Reprinted with permission from Ref. [73]. Copyright 2018 American Chemical Society; (**c**) STEM images of the TCTA/epoxy resin nanocomposites with 1.38 vol % of T_3_C_2_Tx. Reprinted with permission from Ref. [75]. Copyright 2020 AAAS; (**d**) Schematic illustration of possible mechanism of EMI shielding and heat conduction in MXene/AgNWs/ER nanocomposite and SEM image of aerogel with 3 mm thickness. Reprinted with permission from Ref. [56]. Copyright 2022 Elsevier; (**e**) SEM images and schematic illustration of rGH, rGMH, and rGMH/epoxy nanocomposites. Reprinted with permission from Ref. [76]. Copyright 2020 Elsevier.

**Figure 8 polymers-14-01170-f008:**
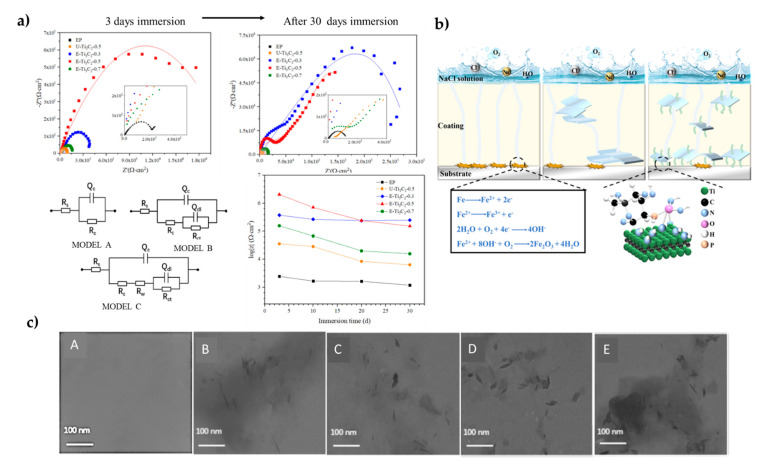
(**a**) Nyquist plots with different immersion times (3 and 30 days) and impedance moduli of neat ER and (GPS)-Ti_3_C_2_T_X_ coatings. Common equivalent circuits to fit EIS data. Model A [R_s_(Q_c_R_c_)] is the equivalent electrical circuit appropriate for describing the initial stage of immersion, and as the immersion time increased, the equivalent circuit is described by Model B [R_s_(Q_c_R_c_(Q_dl_R^ct^), where corrosion gradually penetrates into the epoxy coating, reaching the coating/metal interface. In Model C, the Warburg resistance (R_w_) was included because of the tangential diffusion effect due to lamellar nanofiller addition. Reprinted with permission from Ref. [83]. Copyright 2022 Elsevier; (**b**) Schematic representation of corrosion process of neat epoxy, 0.2 wt % MXene, and 0.2 wt % functionalized MXene coating. Reprinted with permission from Ref. [92]. Copyright 2022 Springer Nature Switzerland; (**c**) TEM images of ultrathin section (**A**–**E**) showing neat ER, unmodified-Ti_3_C_2_Tx-0.5, and epoxy functionalized-Ti_3_C_2_T_x_, epoxy coatings. Reprinted with permission from Ref. [83]. Copyright 2022 Elsevier.

**Table 1 polymers-14-01170-t001:** Comparison of processing methods of MXene/epoxy nanocomposites.

Method		Advantages	Disadvantages
Physical blending	Solvent-assisted blending	▪ Simple and versatile fabrication process ▪ Good dispersion of filler within the resin	▪ Use of organic solvent may impact environmental friendliness and cost ▪ Residual solvent could degrade the properties of nanocomposite ▪ Poor dispersion of filler at high concentrations ▪ Aqueous media easily oxidize MXene
	Mechanical mixing	▪ Avoids the use of solvent ▪ Straightforward performance ▪ Good dispersion. The fillers can achieve highly exfoliated structure	▪ High filler contents are not well-dispersed ▪ Reduction in high aspect ratio of the 2D fillers ▪ High viscosities
Infiltration and impregnation	Infiltration molding	▪ Easy processing, low cost, and minimum waste ▪ Ability to manufacture large complex parts ▪ Filler structure is maintained	▪ Time-consuming process ▪ Difficult to scale up ▪ Inferior mechanical properties
	Vacuum-assisted impregnation		

## Data Availability

Not applicable.

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
