# Peer review of "Recent Advances in MXene/Epoxy Composites: Trends and Prospects"

_polymers, 2022, doi:10.3390/polym14061170_

Round 1
Reviewer 1 Report
In this review, the author overview of the recent progress in the development of MXene/epoxy nanocomposites and the contribution of the nanofiller on the enhancement of properties. Particularly, its application for protective coatings (i.e., anticorrosive and friction and wear), electromagnetic interference shielding, and composites is discussed. Finally, a perspective of the challenges in this topic is presented. This review is very intreating. However, this work can be accepted after addressing the following comments. 1, In order to better understand, the author should add some table about performances compersions.Author Response
Please see the attachment

Reviewer 2 Report
Dear author MXene is a "new approach" with a lot of momentum (relates how fast find also a commercial path). I accepted your article in its current form since I believe you addressed the majority of the relevant topics and aspects.
Reviewer 3 Report
This is very interesting manuscript. Most of the paper focusing MXens in electrical applications not as fillers in composites structure.
Some minor modifications is needed for acceptance:
- Abstract need major revisions. It does not reflect the novelty of MXens in epoxy composites.
- Line 53: But particularly, since.. -> this sentence should be start in a new paragraph.
- In section 2, I would suggest author to highlight the difference MXnes with MWCNT, SWCNT and graphene etc. in term of their structure, properties and performance.
- Section 2.2: what solvent use in physical blending ?
- Please highlight the significant physical and mechanical method to prepare mxens/nanocomposites in Table.
- Which one is correct epoxy/MXene or MXene/ER. Please check and make it consistent.
- Please check, Ti3C2TX.
- Please revise this title, 3.2. Thermal properties and related applications. This is not correct.
- The summary and future outlooks is too long. I advise authors to shorten the summary and future outlooks not more than 3 paragraph. Also please remove the "too general statement".
Reviewer 4 Report
The review proposed by these authors deals with the reinforcement of epoxy resin with MXene to obtain composites with superior properties. The review work is very complete and is worthy of publication. I would suggest to the authors some minor actions to improve the readability. These suggestions are reported in the attached pdf.
